# Proxy-Based Adaptive Transmission of MP-QUIC in Internet-of-Things Environment

Muhammad Hafidh Firmansyah [ID], Joong-Hwa Jung [ID] and Seok-Joo Koh *[ID]

School of Computer Science and Engineering, Kyungpook National University, Daegu 41566, Korea; hafid@knu.ac.kr (M.H.F.); godopu16@gmail.com (J.-H.J.)
* Correspondence: sjkoh@knu.ac.kr; Tel.: +82-53-950-7356

**Abstract:** With the growth of Internet of Things (IoT) services and applications, the efficient transmission of IoT data has been crucially required. The IETF has recently developed the QUIC protocol for UDP-based multiplexed and secure transport. The Multipath QUIC (MP-QUIC) is also being discussed as an extension of QUIC in the multipath network environment. In this paper, we propose a proxy-based adaptive MP-QUIC transmission for throughput enhancement in the IoT environment. In the proposed scheme, a proxy device is employed between IoT clients and IoT server to aggregate the traffics of many clients in the access network. The proxy will transport a large among of traffics to the server, adaptively to the network conditions, by using multiple paths in the backbone network. For this purpose, the proxy device employs a path manager to monitor the current network conditions and a connection manager to manage the MP-QUIC connections with the IoT server over the backbone network with multiple paths. For effective MP-QUIC transmission, the proxy will transmit the prioritized packets to the server using the best path with the lowest round-trip time (RTT), whereas the non-prioritized packets are delivered over the other paths for traffic load balancing in the network. From the testbed experimentations with the MP-QUIC implementation and ns-3 simulation modules, we see that the proposed scheme can outperform the normal QUIC (using a single path) and the existing MP-QUIC scheme (using the round-robin policy) in terms of response delay and total transmission delay. Such performance gaps tend to increase as the link delays and packet loss rates get larger in the network.

**Keywords:** QUIC; MP-QUIC; IoT; proxy; path manager; connection manager

## 1. Introduction

As a lot of Internet of Things (IoT) services and applications have recently been developed, efficient data transmission schemes are crucially required for real-time IoT data traffics. For effective data transmissions over the network, many multipath transport protocols can be used, which include Multi-Path TCP (MP-TCP) [1] and Stream Control Transport Protocol (SCTP) [2]. MP-TCP is an extension of TCP that can utilize multiple paths concurrently in the network to enhance the transmission throughputs. The traditional TCP uses only a single path between the two endpoints, whereas MP-TCP can transfer data by using two or more paths at the same time. SCTP is another transport layer protocol to support the multi-homing features for real-time and high-quality services, such as Internet telephony. With this SCTP multi-homing feature, a more enhanced scheme has also been developed, called Concurrent Multipath Transfer for SCTP (CMT-SCTP) [3]. These MP-TCP and CMT-SCTP schemes can be used for multipath transport effectively, as shown in the related works [4,5]. However, these are basically categorized as 'transport-layer protocols', which are based on TCP and SCTP, usually with *kernel-level* implementations, not *user-level* implementations. Accordingly, there are still some limitations to the wide deployment of these schemes into real-world networks.

In the meantime, the IETF has recently developed a new transport protocol, QUIC, for UDP-based multiplexed and secure transport [6]. The QUIC protocol has been made

based on the previous works on SPDY [7] that are being used for HTTP/2-based Web services. QUIC is a general-purpose transport protocol. It was first designed to support HTTP/3 transmissions [8], but it can also be used for other application protocols. QUIC uses UDP [9] as its underlying protocol, and it provides many distinctive features, which include stream multiplexing, stream, and connection-level flow control, low-latency connection establishment, connection migration, and authentication/encryption with Transport Layer Security version 3 [10]. In particular, it is noted that QUIC can be implemented at the user level. This feature is helpful to easy deployment of QUIC in real-world networks.

More recently, Multi-Path QUIC (MP-QUIC) [11] is being proposed as an extension of QUIC for multipath transport in the network environment with multiple paths. By using MP-QUIC, each data stream of QUIC can be delivered to different paths concurrently in the network. This feature will be helpful to enhance the transmission throughput for mission-critical application services.

This paper addresses how to enhance the throughput performance of MP-QUIC by considering the dynamic network conditions. In this paper, we propose a proxy-based adaptive MP-QUIC transmission in an IoT environment. In the proposed scheme, a proxy device is employed between IoT clients and IoT servers in the network so as to aggregate the data traffics of clients in the access network. The proxy device will be used to transmit these data traffics to the server, adaptively to the dynamic network conditions, by using multiple paths in the backbone network. For this purpose, the proxy employs a *path manager* and a *connection manager*. The *path manager* is used to monitor the current network conditions, and the *connection manager* will be used to manage the MP-QUIC connections between the proxy and the IoT server over the backbone network with multiple paths. For effective MP-QUIC transmission, the proxy will transmit the prioritized packets to the server by using the best path with the lowest RTT, whereas the non-prioritized packets are delivered over the other paths for traffic load balancing in the network.

This paper is organized as follows. Section 2 reviews the existing works on QUIC. Section 3 describes the proposed proxy-based adaptive MP-QUIC transmission scheme. Section 4 discusses the testbed experimentations for performance analysis. Finally, Section 5 concludes this paper.

## 2. Related Works on QUIC and MP-QUIC

QUIC has been designed to solve a number of transport-layer and application-layer problems that have been experienced by modern Web applications. QUIC is similar to TCP + TLS + HTTP/2 but implemented on top of UDP. The key advantages of QUIC over TCP + TLS + HTTP/2 include the connection establishment with low latency, the stream multiplexing to alleviate the head-of-line blocking problem, and the connection migration to solve the rebinding issue of network-address-translator (NAT).

The QUIC has been developed based on the previous works on SPDY [7]. However, the QUIC can provide distinctive features over SPDY in the viewpoint of the connection establishment process and connection migration. During the connection establishment, QUIC can support 0-RTT or 1-RTT data transmissions, whereas the conventional TCP + TLS scheme requires a 3-RTT handshaking process for connection setup. In the meantime, QUIC connections are identified by a 64-bit connection ID. In contrast, TCP connections are identified by a 4-tuple of source address, source port, a destination address, and destination port. This means that if a client changes IP addresses (for example, by moving out of Wi-Fi range and switching over to cellular) or ports (if a NAT box loses and rebinds the port association), any active TCP connections are no longer valid. With the help of the connection migration functionality, the QUIC data transmission can be continued seamlessly, even though the IP address or port number is changed.

The Multipath QUIC (MP-QUIC) has recently been proposed and discussed [12]. MP-QUIC is an extension of QUIC in which a QUIC connection can utilize two or more network paths concurrently for data transmission between client and server. This multipath transmission can be used for traffic load balancing and also throughput enhancement in

the networks. In MP-QUIC, a Path ID is used to identify a specific path in connection with multiple paths. Each client needs to determine the path to be used for data transmission among the candidate paths available in the network. One simple rule to determine the path is a *round-robin* policy in which all available paths will be used for data transmission sequentially in rotation for load balancing.

On the other hand, some works have been made to deploy QUIC into mobile communication networks, such as 3G, LTE, and 5G. The Cross-layer QUIC (C-QUIC) scheme was proposed to use QUIC for next-generation mobile networks [13]. In C-QUIC, the authors propose a variety of policies to determine the network path between cellular link and Wi-Fi link by using the underlying link-layer information, whereas the conventional QUIC uses the 'Wi-Fi First' policy. This approach may be useful to reduce the power consumption of mobile phones and also to enhance the transmission throughput. The authors argue that C-QUIC can provide better performance than the conventional QUIC by 20%, TCP by 36%, MP-TCP by 17%.

Another study is being progressed so as to integrate MP-QUIC into the 3GPP mobile network system, named the Access Traffic Steering, Switching, and Splitting (ATSSS) system [14]. The ATSSS system provides network support for multihomed devices to select a path for transmission (steer), move traffic from one path to another (switch), or use multiple paths simultaneously (split). 3GPP has discussed how to enable ATSSS for non-TCP traffic, based on the use of QUIC.

Until now, we have investigated the existing works for QUIC enhancement, which include MP-QUIC. From the observations, it is noted that there has not been enough works on how to effectively use QUIC or MP-QUIC in the Internet-of-Things (IoT) environment with a lot of IoT devices/sensors. In addition, the works on efficient QUIC transmission by considering dynamic network conditions have not been made enough in the networks where transmission delays or packet loss rates are changed dynamically.

## 3. Proposed Proxy-Based Adaptive MP-QUIC Transmission

### 3.1. System Model and Overall Operations

In this paper, we address how to effectively use the MP-QUIC in the IoT environment that consists of a lot of IoT devices (QUIC clients). To support many IoT clients, we employ a proxy device that is located between clients and the QUIC server. The network environment between client and proxy will act as a local network environment, while the proxy and server can be defined as a large internet environment. The proxy will be used to aggregate data traffics from many clients by using the QUIC connections and then forward the traffics to the server by using an MP-QUIC connection in the backbone network with multiple paths.

More specifically, in this paper, we propose a proxy-based adaptive MP-QUIC transmission scheme. Depending on the dynamic network conditions, the proxy will monitor the current network condition for each available path to the server by measuring the Round-Trip Time (RTTs) periodically. To do this, the proxy employs a *path manager*. To manage the QUIC connections between clients and server, the proxy also employs a *connection manager*, which is used to maintain the mapping relationship between QUIC connections with many clients and the MP-QUIC connection with the server.

Figure 1 shows an overview of the proposed MP-QUIC transmission system. In the figure, many IoT clients communicate with the IoT server by way of the proxy that employs a path manager and a connection manager. There are multiple paths between the proxy and the server. In the proxy section, a bridge of the connection will handle the whole network process. The connection manager and path manager are doing internal communication on the user-space level with an additional program. Because QUIC comes on the user-space programming, it can be easy to edit the primary QUIC system. Our bridges system was built on the QUIC-GO library.

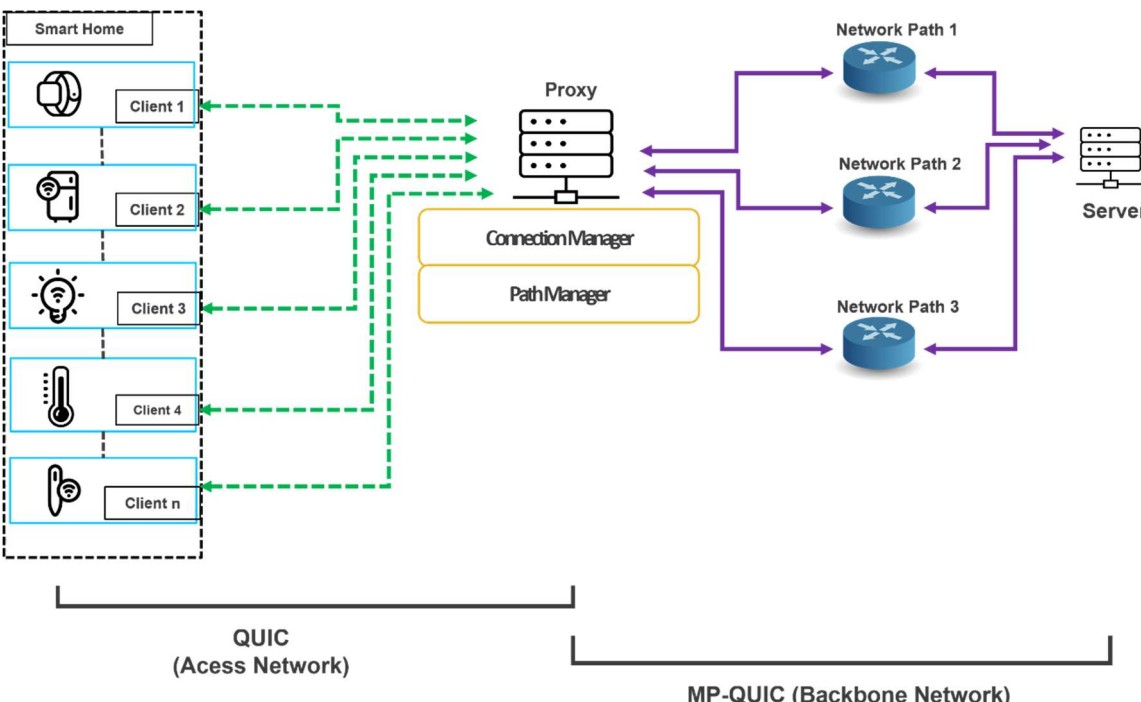

**Figure 1.** Proxy-based MP-QUIC transmission in IoT environment.

In the figure, each client (IoT device) is pre-configured to contact with the proxy, as shown in most of IoT services, such as smart home [15], smart farm [16] and smart factory [17]. The proxy corresponds to the IoT gateway that is used to interconnect IoT clients and IoT server. The access network between clients and proxy is just used to aggregate and forward the traffics of many clients to the proxy. In the access network, IoT clients are connected to the proxy via one or two hops.

The proposed adaptive MP-QUIC transmission scheme is applied mainly to the backbone network between the proxy and the server. It is noted that each client uses the URL of the server for data communication. When the proxy receives the URL of the server who the client wants to communicate with, the proxy will initiate and manage the corresponding MP-QUIC connection with the server by referring to the URL information of the server.

In the proposed scheme, we assume that the clients generate two types of data packets: *priority packets* and *non-priority packets*. The priority packets can be considered as a premium class, and thus these packets will be delivered to the server by using the best path with the smallest RTT among the available network paths. In the meantime, the non-priority packets are regarded as a best-effort class, and thus these packets will be delivered to the server by using a path with the lowest utilization among the available network paths. Accordingly, the probability that a non-priority packet uses the best path will be inversely proportional to the number of priority packets to be delivered. In this paper, we propose the scheme for mapping one connection between clients and proxies to N connections between proxies and servers.

Figure 2 describes the overall operations of the proposed scheme. The proposed proxy-based adaptive MP-QUIC transmission procedures can be divided into three operations: *path management* (by path manager), *connection management* (by connection manager), and *data transmission*. All communications will be done by using HTTP/3 over QUIC.

(1) *Path Management*: The proxy performs the RTT measurements with the server for all available network paths periodically (e.g., 30 s). To do this, the proxy sends a *Path Request* to the server (Step 2). The server then responds with *ACK Response Path* to the proxy (Step 3). The proxy can now calculate and update the RTT value for the path.

(2) *Connection Management*: A client who wants to communicate with the server will first contact the proxy (Step 4). For this purpose, it is assumed that each IoT client already knows the contact information of the proxy by pre-configuration. The client transmits a GET message that contains the Hash ID request (this will be described later) and the priority type (0: non-priority, 1: priority) to the proxy. Then, the proxy creates a Hash ID for the client (Step 5) and sends this information to the client (Step 6). After that, the proxy will determine the MP-QUIC path to the server for this client by internally contacting the path manager (Step 7~9).

(3) *Data Transmission*: After getting the Hash ID from the proxy, the client can now transmit the data packets. Each data packet should contain the Hash ID information (Step 10). When the proxy receives the data packet from the client, it can determine the MP-QUIC network path for the client, and thus it will update the '*hits*' information for the path (Step 11). This *hits* information will be referred to for the MP-QUIC transmission of the non-priority packets later, which will be described in the subsequent section. Now, the proxy will forward the data packet to the server (Step 12), and the data communication will be done between the client and the server by way of the proxy (Step 13).

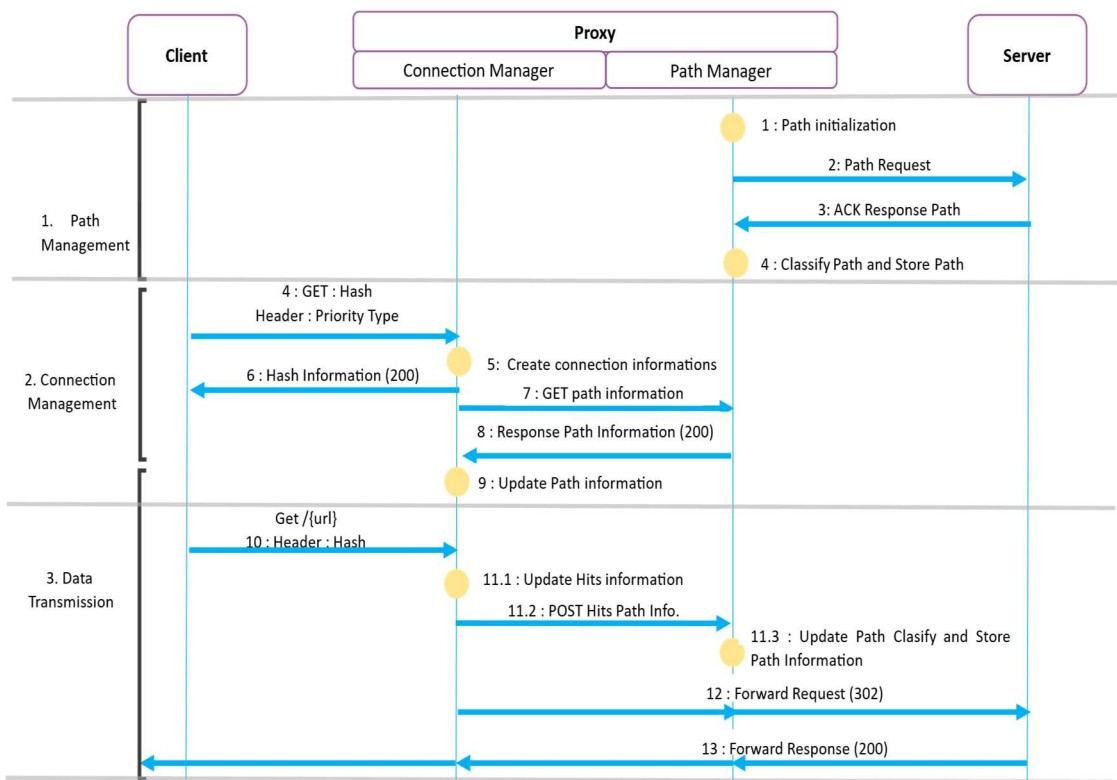

**Figure 2.** The overall operation of the proposed MP-QUIC transmission.

### 3.2. Path Management

The proxy employs the path manager that is used to monitor each available path between proxy and server. For this purpose, the proxy measures the RTT for each path by exchanging Path Request and Response messages with the server periodically.

From these RTT measurements with the server, the proxy will maintain the path management table, as shown in Table 1. In the table, RTT represents the RTT value that has

been measured and updated for each available path (or IP address). On the other hand, the '*hits*' represent the number of clients who are using this network path. This value will be referred to and updated in the connection management operation. The QUIC connection of the client with the same priority packets will use the network path with the lowest *hits* value.

**Table 1.** Path management table.

| Path ID | RTT (ms) | Hits |
|---|---|---|
| Path_1 | 100 | 10 |
| Path_2 | 200 | 4 |
| Path_3 | 300 | 2 |
| Number of total connections | | 16 |

In the path management operations, the proxy monitors all candidate paths to the server in the backbone network by measuring the RTT for each candidate path 'periodically' (e.g., 30 s). This is because the measured RTT values can vary over time, and thus we continue to measure the recent RTT values periodically. Among many candidate paths, the proxy will choose the best path with the smallest RTT as the data transmission path.

*3.3. Connection Management*

In the proposed scheme, the connection manager is used to manage all connection requests by clients and to determine the MP-QUIC for each client by considering the concerned priority. The connection manager uses the Hash ID to identify each client, as shown in Table 2.

**Table 2.** Connection management table by Hash ID.

| Hash ID | Client | | Server | | Priority |
|---|---|---|---|---|---|
| | IP Address | Port Number | IP Address | Port Number | |
| 123ABC | 1.1.1.1 | 5389 | 10.10.10.10 | 443 | 0 |
| 456DEF | 1.1.1.2 | 4573 | 10.10.10.5 | 443 | 1 |
| 789GHI | 1.1.1.3 | 5482 | 10.10.10.4 | 443 | 0 |
| ... | ... | ... | ... | ... | ... |
| 754KCH | 1.1.1.10 | 5678 | 10.10.10.5 | 443 | 1 |

Hash ID for a client is allocated by the proxy. For each Hash ID, the mapping information between client (QUIC) and server (MP-QUIC) will be maintained and updated, which includes the IP address and port number of the client, the IP address, and port number of the server, and the concerned priority type.

In this paper, we assume that each client determines its priority (0 or 1), as per the pre-configuration and an appropriate negotiation with the proxy (or service provider). In Step 4 of Figure 2, a client requests a Hash ID to the proxy with its priority information. In Step 5, the proxy creates a new entry for the client in the connection management table, in which the priority information is recorded, as shown in Table 2. In Step 6, the server responds to the client with Hash ID information.

In the connection management, the proxy will determine the MP-QUIC path for the client among all available network paths, as indicated in the path management table of Table 1. The client with priority packets will be assigned to the best path with the smallest RTT among available MP-QUIC paths between the proxy and the server. On the other hand, the client with non-priority packets will be assigned to the path with the lowest *hits* value for traffic load balancing among the available MP-QUIC paths while it is available. If the two MP-QUIC paths have the same *hits* value, the path with a smaller RTT is preferred. In Step 6 and 7, the connection manager determines this MP-QUIC path information by interworking with the path manager.

*3.4. Data Transmission*

When a client sends data packets, the proxy performs the data transmission operations, as described in Step 10~Step 13 of Figure 2. The data transmission between the proxy and the server will be performed adaptively to the current network conditions, based on the information obtained by the path management and connection management.

In Step 10 of Figure 2, a client sends an HTTP/3 packet to the proxy by using QUIC. The HTTP/3 header will include the Hash ID information that was assigned by the proxy (Step 6 of Figure 2). By referring to the connection management table (Table 2), the proxy can determine the MP-QUIC path that will be used for data forwarding to the server, which depends on the priority type.

Once the MP-QUIC path is determined, the proxy (connection manager) increases the Hits values in the connection management table, as described in Step 11.1~Step 11.3 of Figure 2. Then, the data packets will be forwarded to the server by using the determined MP-QUIC path (Step 12 of Figure 2). In turn, the server responds with the data packets to the client by way of the proxy.

The remaining data transmissions between the client and the server via the proxy will be performed in a similar way, as described until now. During data communication, the tables for path management and connection management will continue to be updated and maintained for further subsequent data transmission of the new clients.

## 4. Performance Analysis by Experimentations

*4.1. Testbed Configuration*

This section discusses the testbed configuration for performance analysis of the proposed scheme. All experiments were performed by using the *network simulator* version 3 (*ns-3*) [18], the MP-QUIC library [11], and the *docker* [19]. The docker has been implemented by using the DOCKEMU [20,21].

For performance analysis, we compare the proposed proxy-based adaptive MP-QUIC transmission with the existing two QUIC schemes: (1) SP-QUIC using a single path [22] and (2) the conventional MP-QUIC [11] that uses a simple *round-robin* policy to determine the MP-QUIC path among available network paths, in which all available paths are used for data transmission sequentially in rotation. This performance evaluation is mainly targeted to the multipath network between the proxy and the server. As for performance metrics, we employ the *response delays* from the server and the *total transmission delays* required for transmission of 128~1024 KB data files by using HTTP/3 over QUIC. Each packet contains the payload of 50 bytes. To analyze the performance impacts on a variety of network conditions, we performed experiments for different link delays and packet loss rates in the network.

To configure the multipath network environment, we employ the three different network paths between proxy and server, as shown in Figure 1. The number of clients (*n*) is set to 10 clients, and extended up to 50 clients. Each client sends both priority types, and the network bandwidth for each network link is set to 100 Mbps, 75 Mbps, 50 Mbps for Path-1, Path-2, and Path-3, respectively. The default value of link delay is set to 10 ms for each path in the backnbone network.

*4.2. Analysis of Response Delays*

Figure 3 shows the comparison of response delays from the server to the client of the three candidate schemes for different link delays. The link delays for each path are changed from 10 ms to 50 ms. From the figure, we see that the proposed scheme provides smaller response delays than the two existing schemes. As the link delay gets larger, the response delay tends to increase. The two existing schemes, SP-QUIC (using single path) and MP-QUIC (Round Robin), provide similar and larger response delays than the proposed scheme. This is because the proposed adaptive MP-QUIC scheme can utilize the available link for data transmission adaptively the network conditions. Using the proposed scheme can keep the response delay remains lowest, this can happen when link delay

increase, proxy system migrate the connection using path manager. While another scheme is not concerned with adaptive path changing, the proposed scheme can provide the best path for the proposed MP-QUIC.

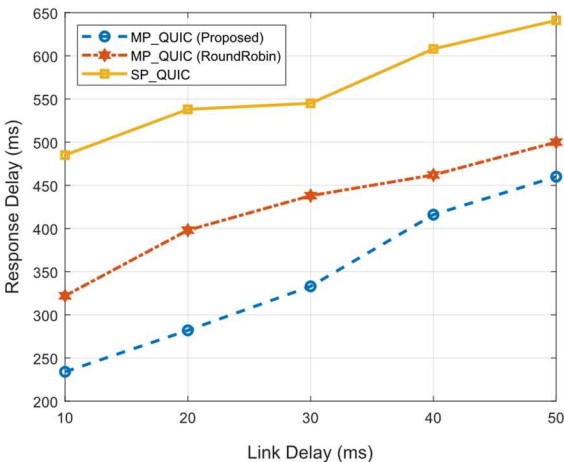

**Figure 3.** Comparison of response delays for different link delays.

Figure 4 shows the response delays of the three candidate schemes for different packet loss rates in the network, which are ranged from 0% to 15%. In the figure, the response delays tend to increase as the packet loss rate gets larger. We see that the proposed adaptive MP-QUIC transmission scheme provides smaller response delays than the existing SP-QUIC and MP-QUIC schemes. This is because the proposed adaptive scheme can transmit the data packets over the best path among the multiple paths by considering the network conditions, including the packet loss rate. In the figure, we can also see that the performance gaps between the proposed scheme and the existing schemes tend to increase as the packet loss rate gets larger.

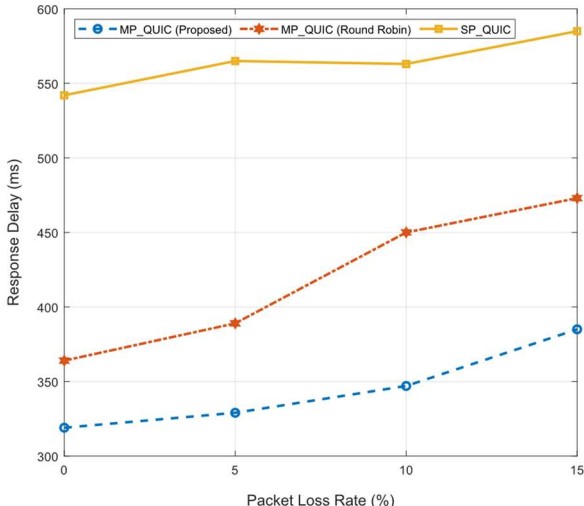

**Figure 4.** Comparison of response delays for different packet loss rates.

Figure 5 compares the response delays of the candidate scheme for the different sizes of data files transmitted, which are ranged from 128 KB to 1024 KB. From the figure, we see that the proposed scheme provides the best performance among the three candidate schemes for all test instances.

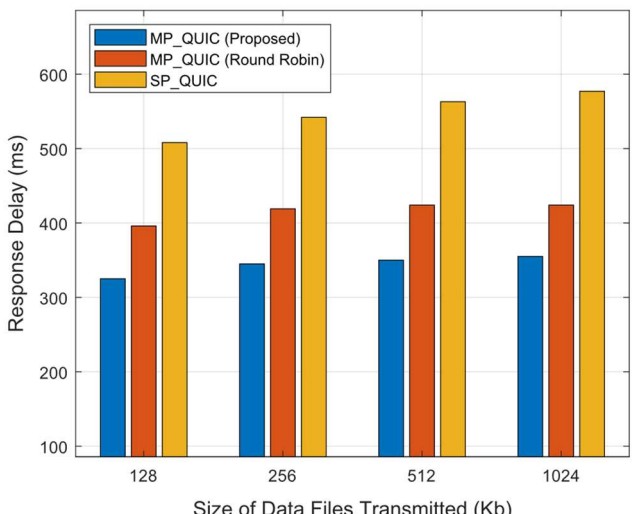

**Figure 5.** Comparison of response delays for different sizes of data files transmitted.

### 4.3. Analysis of Total Transmission Delays

In this section, we analyze the performance of total transmission delays for three candidate schemes. The total transmission delay is measured by the time that has been elapsed to complete the transmission of all data files from clients to the server.

Figure 6 shows the total transmission delays of the three candidate schemes for different link delays. The link delays for each path are changed from 10 ms to 50 ms. In the figure, we see that the proposed adaptive MP-QUIC scheme provides smaller response delays than the existing SP-QUIC and MP-QUIC schemes. This is because the proposed scheme can utilize dynamic network information for data transmission adaptively to the current network conditions.

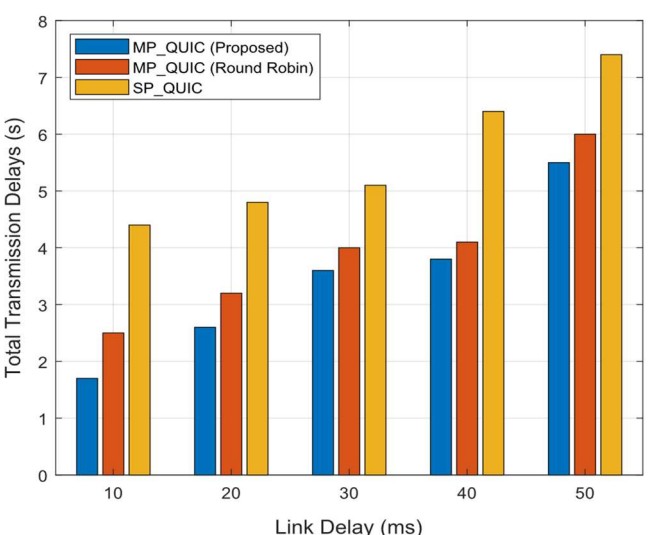

**Figure 6.** Total transmission delays for different link delays.

Figure 7 shows the total transmission delays of the three candidate schemes for different packet loss rates, ranging from 0% to 15%. We see that the proposed adaptive MP-QUIC transmission scheme provides smaller response delays than the existing SP-QUIC and MP-QUIC schemes. This is because the proposed adaptive scheme can transmit the data packets over the best path among the multiple paths by considering the network conditions, including the packet loss rate.

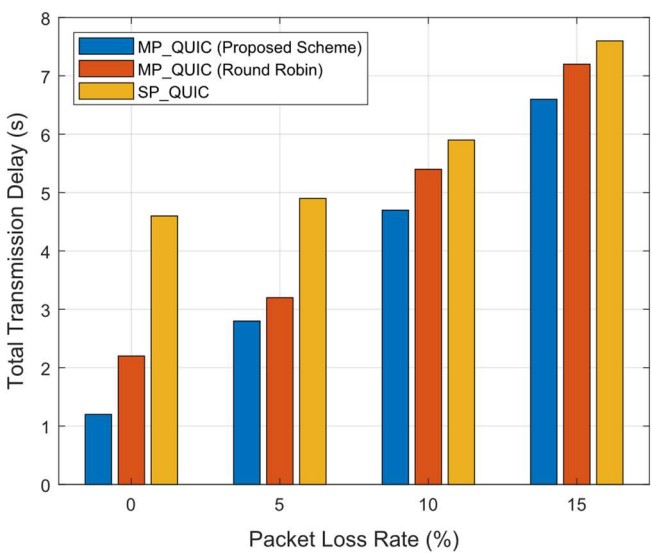

**Figure 7.** Total transmission delays for different packet loss rates.

Figure 8 shows the comparison of total transmission delays for different data file sizes, which are ranged from 128 KB to 1024 KB. From the figure, we see that the proposed scheme provides the best performance among the candidate schemes for all test instances. The gaps of transmission delays between the proposed scheme and the existing schemes get larger as the data file size increases.

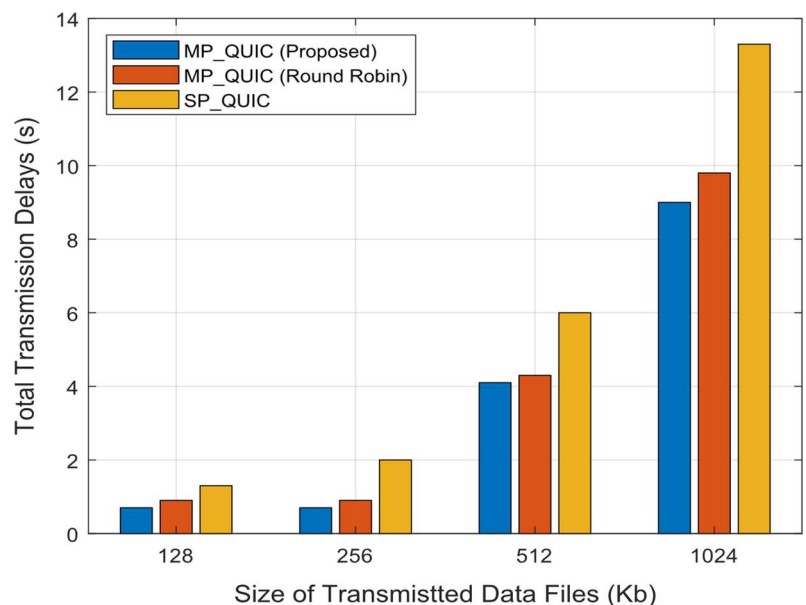

**Figure 8.** Total transmission delays for different sizes of data files transmitted.

### 4.4. Analysis of Packet Priority and Number of Clients

The proposed adaptive MP-QUIC is performed by considering the priority of data packet. To analyze the impacts of packet priority on performance, we compare response dalays for candidate schemes when all data packet have the same priority (priority = 1), as shown in Figure 9. From the results, we see that the existing SP-QUIC gives the worst performance. The proposed adaptive MP-QUIC scheme provides the best performance among all candidate schemes. This is because in the proposed scheme the candidate path with a lower hit value or a lower RTT value will be preferred for data transmission. This tends to induce traffic load balancing and performance improvement.

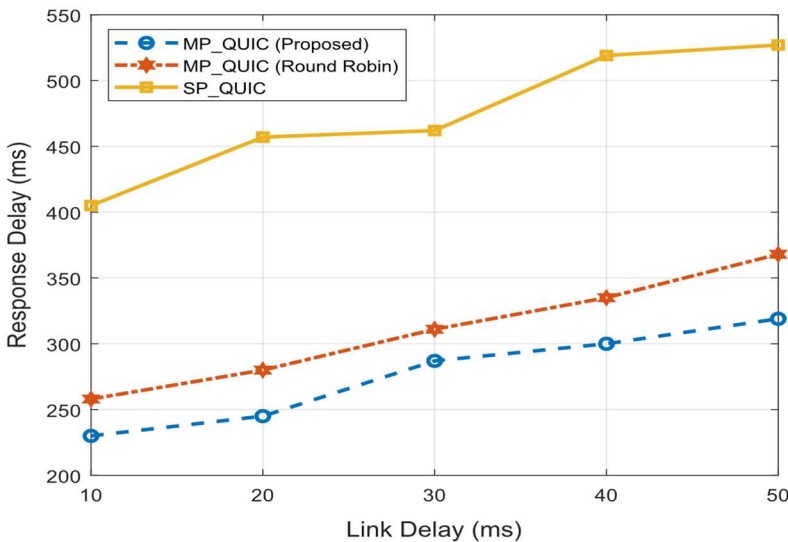

**Figure 9.** Response delays for different link delays when all pakcets have the same priority.

Figure 10 shows the performance comparison for experimens with different numbers of clients, in which the number of clients is ranged from 10 clients to 50 clients. From the results, we see that the proposed adaptive MP-QUIC scheme still provides the best performance. This is because in the proposed scheme the proxy tries to keep the response delays as low as possible by using adpative transmission and packet priority, even when the number of clients gets larger in the network. In the meantime, in the existing SP-QUIC and MP-QUIC schemes, the response delays tend to get larger, as the number of clients increases.

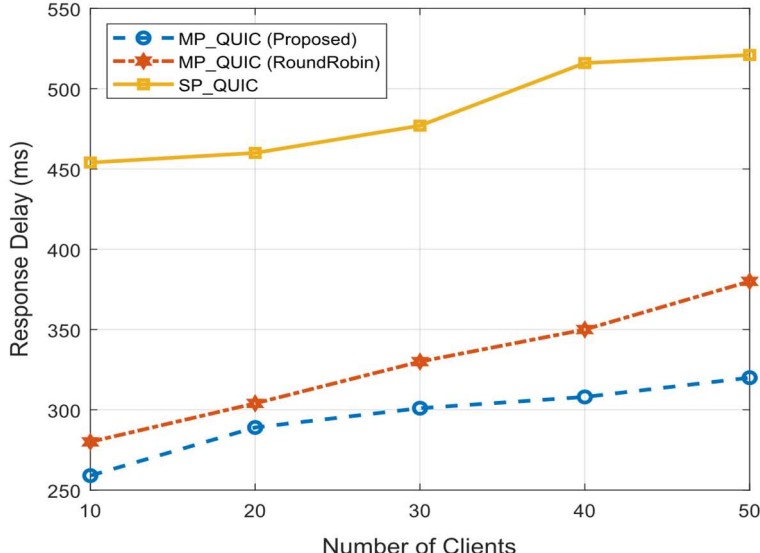

**Figure 10.** Comparison of response delays for different number of clients.

## 5. Discussion

This paper has proposed an enhanced MP-QUIC scheme for throughput enhancement in an IoT environment. In the proposed scheme, a proxy device is employed between IoT clients and IoT servers to aggregate the traffics of many clients in the access network. The proxy device transmits aggregated IoT traffics to the server, based on dynamic network conditions, by using multiple paths in the backbone network. The proposed scheme is also featured by adaptive MP-QUIC transmission, in which the proxy employs a path

manager to monitor the current network conditions, and a connection manager to manage the MP-QUIC connections with the IoT server over the backbone network with multiple paths. In MP-QUIC data transmissions, the prioritized packets are delivered to the server by using the path with the lowest network latency, whereas the non-prioritized packets are delivered over the paths with low utilization for traffic load balancing in the network.

It is noted that the Social IoT (SIoT) services have recently been studied [23–25]. SIoT can be used to interconnect many IoT objects over the world and to create a social network based on common interests and motivation. SIoT addresses how to effectively combine social networks with IoT, which is related to the interaction between things and the Internet as a network substrate [23]. For this purpose, a variety of technical issues have so far been discussed [24]. To support a variety of SIoT applications, the different software platforms from the conventional IoT platforms may need to be considered by utilizing the SIoT features [25].

The proposed proxy-based MP-QUIC scheme can be integrated with the SIoT architecture so as to more effectively provide the SIoT services. In general, a large number of IoT devices are involved into the SIoT services, and these IoT devices can be categorized into several groups, based on their geographical locations and common interests. Each group can be assigned to a specific MP-QUIC proxy, and the proxy will communicate with a different proxy or the central SIoT server by using the proposed MP-QUIC scheme so as to enhance the transmission throughput. That is, the proposed proxy-based MP-QUIC scheme will be suitable to provide the scalable SIoT data delivery.

More recently, the Multiple IoTs (MIoT) paradigm has been proposed as an extension of SIoT [26–28]. MIoT is a set of inter-related SIoT networks. In [26], the authors investigated the possibility of applying the ideas underlying Social Internetworking System to IoT, and they proposed the MIoT paradigm. MIoT addresses data-driven and semantics-based aspects because it considers the contents exchanged by smart objects during their transactions. Under the MIoT definition, SIoT can be viewed as a specific case of MIoT in which the number of the possible kinds of relationship is limited. SIoTs are interconnected via the so-called *cross nodes*. Then, a *cross node* connects at least two SIoTs of the MIoT. In [27], the authors define the concept of *scope* in MIoT, and they propose the formalizations of this concept. In [28], the authors propose a theoretical framework to handle anomalies in multiple IoT scenarios with the definition of anomaly taxonomies.

The proposed proxy-based adaptive MP-QUIC transmission scheme can also be integrated into the MIoT framework, in which an SIoT consists of a set of IoT objects with the common interests, and these IoT objects are connected to the MP-QUIC proxy. The MP-QUIC proxy can be regarded as an MIoT *cross node* which will communicate with the other MP-QUIC proxy that represents another SIoT group. In this way, many SIoTs can be inter-connected via MP-QUIC proxies in the MIoT framework, and the proposed adaptive MP-QUIC transmission scheme can be used to provide real-time and reliable data transmission among the MP-QUIC proxies in the network.

## 6. Conclusions

In this paper, we described a proxy-based adaptive MP-QUIC transmission for throughput enhancement in the IoT environment. In the proposed scheme, a proxy device is employed between IoT clients and IoT server to aggregate the traffics of many clients in the access network. From the testbed experimentations, we can see that the proposed scheme outperforms the normal QUIC (using a single path) and the existing MP-QUIC scheme (using the round-robin policy) in terms of response delay and total transmission delay. This is because the proposed adaptive scheme can transmit data packets over the best path with lower RTTs and/or lower utilization among the multiple paths by considering the network conditions, such as link delay and packet loss rate. The proposed proxy-based adapative MP-QUIC transmission scheme tends to give higher throughput performance than the existing QUIC schemes.

In the proposed scheme, many IoT clients are connected to the pre-configured proxy. This can be regarded as a centralized scenario, which may be subject to the issue of a single point of failure. For furthey study, the distributed IoT scenario can be considered as a more elaborated scheme to handle a single point of failure.

**Author Contributions:** M.H.F. wrote the initial manuscript; J.-H.J. revised the manuscript; S.-J.K. proofread the manuscript; All authors have read and agreed to the published version of the manuscript.

**Funding:** This research was supported by Basic Science Research Program through the National Research Foundation of Korea (NRF) funded by the Ministry of Education (NRF-2021R1I1A3057509).

**Conflicts of Interest:** The authors declare no conflict of interest.

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
