# Peer review of "Proxy-Based Adaptive Transmission of MP-QUIC in Internet-of-Things Environment"

_electronics, doi:10.3390/electronics10172175_

Round 1
Reviewer 1 Report
The authors introduce a proxy between IoT devices and a backend server, where the MP-QUIC protocol is employed between proxy and server and QUIC between IoT devices and proxy.
Since the proxy has knowledge about the connection requests from the IoT devices on the one hand and the available paths between the proxy and the server, it can optimize the transmission between proxy and server using a connection manager and a link manager. This is also reflected in the presented results. To that extend the paper is describing the approach in an appropriate form.
However, it falls short in several respects:
- The paper assumes that all IoT devices are pre-configured to know the respective proxy. This doesn't allow for dynamic network configuration as may be very relevant in IoT scenarios.
- Two priority classes are assumed. However, the paper doesn't analyze the impact of the priority distribution (e.g. the corner cases of all requests with same priority).
- The proxy concept essentially centralizes the communication, which may be a unrealistic assumption in a distributed IoT scenario (+ it establishes a single point of failure).
- For the performance analysis, no other proxy based scheme is tested (which certainly exists).
- After introducing the proxy concept to manage the "lots of IoT devices" the analysis is run with only 5 IoT devices, which is not representative. It is unclear how the proxy concept performs under large IoT device load.
- The data sizes are very large for IoT devices with 10s of MBs, the performance should rather be evaluated with small packet sizes for many devices.
- The hop (and the associated variable channel) between IoT devices and proxy is not taken into account (or not properly described).
Overall, even though the paper is well written (accept for some English issues), the analysis of the proposed scheme is insufficient.
Reviewer 2 Report
1. Paper talks about implementation of connection manager which is based on hash ID for each connection coming from the client. 2. For each connection, already marked with priority, will be assigned the path determined by path manager. 3. Taking advantage of existing path manager, to mark multi paths to server based on minimal RTT and number of hits. 4. Assign the path based on the point determined in 3. Concerns and suggestions: The overall idea of path selection based on RTT is useful. But is has been widely used. The paper does not present any novel methods. In figure 1, how does the proxy know to what servers it must establish connection with? The client could be contacting any server on the internet! Proxy can establish and measure connection quality only when the client has requested for a connection. The biggest and most notable shortcoming is the lack of information about path management: How does the proxy system ensure that a RTT over a path remains the same? How do you ensure future packets are transmitted over the same path? we have no control over paths. The papers assumes internet core, so there is no control over paths. This questions the validity of the whole work. For data analysis and plotting, please use tool such as Python matplotlib or Matlab instead of Office. Also, figure must be exported in vector PDF format. They are doing QUIC with HTTP3 on client side of the proxy, but unclear how client and server side are matched. If they are claiming merely based on the hash id they calculate then it is wrong because hash ID can change based on 5-tuple but QUIC connection be still same due to same CID. If client -> proxy is QUIC and proxy to server is Multipath QUIC then how the integration of these two work, is unclear.Author Response
Please see the attachment.

Round 2
Reviewer 1 Report
The authors have addressed most of my comments and concerns appropriately and the revised results look meaningful to me.
Therefore I agree with the publication of the revised paper.
Reviewer 2 Report
My main concern is that the paper does not provide any significant contributions. The proposed work is very simple. No novel algorithm or method has been proposed.
It is simply the performance evaluation of MP-QUIC in a particular scenario. The paper is more suited for a conference publication.
